# The Association between Muscle Quantity and Overall Survival Depends on Muscle Radiodensity: A Cohort Study in Non-Small-Cell Lung Cancer Patients

**DOI:** 10.3390/jpm12071191

**Published:** 2022-07-21

**Authors:** Wouter A. C. van Amsterdam, Netanja I. Harlianto, Joost J. C. Verhoeff, Pim Moeskops, Pim A. de Jong, Tim Leiner

**Affiliations:** 1Department of Radiology, University Medical Center Utrecht, Utrecht University, Heidelberglaan 100, 3584 CX Utrecht, The Netherlands; n.i.harlianto@umcutrecht.nl (N.I.H.); p.dejong-8@umcutrecht.nl (P.A.d.J.); timleiner@gmail.com (T.L.); 2Babylon Health, 1 Knightsbridge Green, London SW1X 7QA, UK; 3Department or Radiation Oncology, University Medical Center Utrecht, Utrecht University, Heidelberglaan 100, 3584 CX Utrecht, The Netherlands; j.j.c.verhoeff-10@umcutrecht.nl; 4Quantib BV, Westblaak 106, 3012 KM Rotterdam, The Netherlands; p.moeskops@quantib.com; 5Department of Radiology, Mayo Clinic, 200 First St. SW, Rochester, MN 55905, USA

**Keywords:** cachexia, carcinoma, non-small-cell lung, survival analysis, prognosis

## Abstract

The prognostic value of CT-derived muscle quantity for overall survival (OS) in patients with non-small-cell lung cancer (NSCLC) is uncertain due to conflicting evidence. We hypothesize that increased muscle quantity is associated with better OS in patients with normal muscle radiodensity but not in patients with fatty degeneration of muscle tissue and low muscle radiodensity. We performed an observational cohort study in NSCLC patients treated with radiotherapy. A deep learning algorithm was used to measure muscle quantity as psoas muscle index (PMI) and psoas muscle radiodensity (PMD) on computed tomography. The potential interaction between PMI and PMD for OS was investigated using Cox proportional-hazards regression. Baseline adjustment variables were age, sex, histology, performance score and body mass index. We investigated non-linear effects of continuous variables and imputed missing values using multiple imputation. We included 2840 patients and observed 1975 deaths in 5903 patient years. The average age was 68.9 years (standard deviation 10.4, range 32 to 96) and 1692 patients (59.6%) were male. PMI was more positively associated with OS for higher values of PMD (hazard ratio for interaction 0.915; 95% confidence interval 0.861–0.972; *p*-value 0.004). We found evidence that high muscle quantity is associated with better OS when muscle radiodensity is higher, in a large cohort of NSCLC patients treated with radiotherapy. Future studies on the association between muscle status and OS should accommodate this interaction in their analysis for more accurate and more generalizable results.

## 1. Introduction

Skeletal muscle quantity is related to patient prognosis in non-small-cell lung cancer (NSCLC) for several reasons. First, muscle loss occurs more frequently in patients with aggressive tumors with high catabolic activity [1]. Second, patients who have low muscle mass are less capable of enduring intensive anti-cancer treatments such as surgery [2], chemotherapy [3] and radiotherapy [4].

The psoas muscle index (PMI) and the skeletal muscle index (SMI) are standardized measurements of muscle quantity on computed tomography (CT) scans. PMI is defined as the cross-sectional area of the psoas muscle on the CT slice corresponding to lumbar vertebra 3 (L3), divided by the square of the height of a patient [5]. SMI is analogously defined by taking the cross-sectional area of all skeletal muscle on the L3 slice instead of only the psoas muscle. Studies in NSCLC patients report that higher muscle quantity correlates with improved overall survival (OS) [6,7,8], improved progression-free survival [9] and better response to treatment [10]. However, most of these results represent univariable associations and several studies did not find that muscle quantity was correlated with these outcomes [1,11,12,13]. In addition to muscle quantity, the radiodensity of muscle tissue has gained interest as a potential prognostic marker in NSCLC [11,12,13,14,15,16,17]. Pro-inflammatory cytokines are elevated in cancer patients and lead to fatty infiltration of muscle tissue. Fatty infiltration can be measured on a CT scan, as fat tissue has a lower radiodensity than muscle tissue. Standardized measurements of fatty muscle infiltration are the average radiodensity in the psoas muscle area on the L3 level, known as psoas muscle radiodensity (PMD), and the analogous average radiodensity in all skeletal muscle on L3, known as skeletal muscle radiodensity (SMD).

PMD and SMD provide measurements of the extent of fatty infiltration of muscle tissue. When muscle tissue is replaced by intra-muscular adipose tissue, muscle strength decreases. We therefore hypothesize that in patients with high muscle radiodensity, muscle quantity is more positively associated with OS than in patients with low muscle radiodensity. This would imply that there is a statistical interaction between muscle quantity and muscle radiodensity for OS. Whereas there is a large body of research on the prognostic value of muscle quantity and muscle radiodensity for OS in lung cancer separately (see for example these systematic reviews: [18,19]), few studies have investigated the associations of both muscle quantity and muscle radiodensity with OS [11,12,13,14,15,16,17]. Whether the association between muscle quantity and OS depends on muscle radiodensity has not been studied before. If the association between muscle quantity and OS indeed depends on muscle radiodensity, the apparent association between muscle quantity and OS will differ across studies, if their study populations differ in distribution of muscle radiodensity. If our hypothesis is true, OS prediction models that include the interaction between muscle quantity and muscle radiodensity will be more accurate and more stable across populations. Therefore, we investigated whether there is a statistical interaction between muscle quantity and muscle radiodensity for OS prediction in a large cohort of NSCLC patients treated with radiotherapy.

## 2. Materials and Methods

### 2.1. Data Source

We conducted a retrospective observational cohort study at the department of radiotherapy of the University Medical Center Utrecht. Patients were referred to our center from 9 different hospitals in the Utrecht province, the Netherlands. This study was conducted in accordance with the applicable privacy guidelines and the declaration of Helsinki and its later amendments. As this was a retrospective study and most of the patients had died, we obtained a waiver for informed consent from the institutional review board at the University Medical Center Utrecht (reference number WAG/mb/19/005583).

### 2.2. Patient Inclusion

We identified and included patients aged 18 years or above when they first visited the radiotherapy department for consideration of treatment with radiotherapy for NSCLC between January 2009 and September 2018. Some patients had multiple episodes of NSCLC for which they received radiotherapy. For these patients, we only included the first episode. There were no exclusion criteria, meaning that all patients were included in the final analysis. Patients who visited the radiotherapy department but for some reason did not receive radiotherapy were not excluded because they are part of the population in which a hypothetical outcome prediction model would be used. There was no minimum follow-up time.

### 2.3. Definition of Determinants and Outcome

Clinical variables were extracted from electronic health records (EHR). The outcome for this study was OS measured on a continuous time scale. The start of follow-up was the date of the first visit to the radiotherapy department. This is generally when treatment decisions are made and where prognostic models have the highest potential impact. If a date of death was not registered in the health records, the Dutch Personal Records Database was queried to verify survival status. The last date of follow-up was 26 April 2021. We made sure to record clinical variables as they were known at the start of the follow-up.

Patients were staged according to the American Joint Committee on Cancer Tumor-Node-Metastasis (TNM) staging protocol. We maintained the TNM version that was clinically used at the time of treatment, spanning versions six [20], seven [21]. Positron Emission Tomography–Computed Tomography (PET-CT) scans made within the time window of 90 days before follow-up start to 30 days after follow-up start were used for the body composition measurements. PET-CT scans were identified in the digital picture archiving and communication system from the radiotherapy department. We used PET-CT scans for our measurements as thoracic CT scans generally do not contain the required L3 level. If there were multiple eligible PET-CT scans for a patient, the scan that was closest to the start of follow-up was used. The PET-CT scans were made over a nine-year period and in nine different hospitals, which means there was natural variation in scanner vendor, model, tube current, voxel spacing, slice thickness and radiation dose. We used Quantib Body Composition version 0.2.1 (Quantib BV, Rotterdam, The Netherlands) for automated muscle measurements [22]. The CT scans were first resampled to a uniform slice thickness of 5mm. The first step for the algorithm was to select the CT slice in the middle of the third lumbar vertebra. On this slice, as well as the two slices above and the two slices below this center slice, the psoas muscle tissue was automatically segmented bilaterally. The cross-sectional area of the segmentation was measured in centimeters squared, and subsequently averaged over the five segmented slices covering a range of 2.5 cm. To calculate the psoas muscle area, only voxels with a radiodensity of −30 Hounsfield Units (HU) or higher were counted to exclude intra-muscular fatty tissue. The psoas muscle area was divided by the square of the height of a patient measured in meters to obtain the PMI. As a second measurement, the PMD was measured as the mean HU value in the entire segmented region, including voxels with radiodensity lower than −30 HU [12,16]. The definitions of PMI and PMD are illustrated in Figure 1. PET-CT scans are rarely acquired with intravenous (IV) iodinated contrast, but if a scan was obtained after IV contrast injection, only the PMI measurement was used, as iodinated contrast artificially increases radiodensity of muscle tissue. All automated segmentations were verified for correctness in joint reading sessions by three experienced readers (WA, NH, TL) including a board-certified radiologist with over 15 years of clinical experience (TL). If the automated segmentation failed, it was corrected manually by one of the authors (NH). The process of creation, verification and correction of segmentations was blinded to the outcome of OS and other patient information.

The following baseline clinical characteristics were extracted from the EHR: age, sex, histology group (grouped as adenocarcinoma, squamous cell carcinoma, no histology obtained, or other), performance score defined by the Eastern Cooperative Oncology Group [23] and body mass index (BMI), defined as patient weight in kilograms divided by the square of their height in meters. These variables were selected based on their wide availability in clinical practice and their frequent inclusion in prognostic models. As patients may lose weight because of their NSCLC, the time window for weight measurements was 90 days before to 30 days after the start of follow-up.

## 3. Statistical Analysis

### 3.1. Model Definition

We used Cox proportional-hazards regression to model OS. As the purpose of this study was to evaluate the potential added prognostic value of the interaction between PMI and PMD, the baseline clinical covariates were included in all tested models. Before entering the analysis, we subtracted the mean of each continuous variable so that they have a mean of 0. Potential non-linear effects of the continuous predictors (age, BMI, PMI and PMD) were investigated by including restricted cubic spline terms using 5 knots, which leads to 4 degrees of freedom per variable. The potential additive interaction on the log-hazard ratio scale between PMI and PMD was investigated by including interaction terms. Only interaction terms that were linear in either PMI or PMD were included, to reduce the degrees of freedom needed to model the interaction. The average OS between different clinical disease stages is very different and the proportional hazards assumption is unlikely to hold for clinical stage. Therefore, the baseline hazard function was stratified per clinical stage in four groups (I, II, III and IV). The patient selection mechanism for radiotherapy is different for early-stage (I and II) and advanced-stage (III and IV) NSCLC. For early-stage NSCLC, patients without contra-indications for surgery are recommended for surgical treatment [24,25]. For stage III, radiotherapy is a standard part of treatment [24,25]. For stage IV, potential indications for radiotherapy are aggressive local treatment of oligometastastatic disease [24] or palliative care on a case-by-case basis [24]. As in early-stage NSCLC the treatment selection is dependent on fitness for surgery, the treatment choice is likely correlated with muscle quantity and radiodensity. As the patient selection mechanism differs between early-stage and advanced-stage, we stratified the hazard ratios per early-stage (stages I and II) and advanced-stage (stages III and IV). The full model included 58 parameters in total.

### 3.2. Sample Size Calculation

We used simulations to calculate the power needed to detect a hazard ratio of 0.986 for a linear interaction term between PMI and PMD for several different sample sizes and correlations between covariates. The assumptions for the sample size calculations were based on three published studies [12,14,17]. The simulations indicated that 1000 patients were sufficient for a power of 0.8 using a two-sided Student’s *t*-test with alpha = 0.05 for a wide range of correlations between variables. The Appendix A presents a detailed report on the assumptions and results of the sample size calculation.

### 3.3. Missing Data

The presence or absence of a PET-CT scan for a patient in our study depends on medical decisions made during the diagnostic and treatment planning process. It is likely that these decisions are correlated with the clinical variables under study and the outcome OS. This means that excluding all patients with missing data (‘complete-case analysis’) would lead to biased parameter estimates [26]. Given the baseline clinical variables included in our study and the outcome OS, the assumption of missing at random conditional on these variables may be tenable. In this situation, multiple imputation yields unbiased parameter estimates and increases the statistical power as more patients are included in the analysis [27]. Therefore, missing data in both the baseline clinical covariates and the scan-derived muscle measurements PMI and PMD were imputed using multiple imputation. To accommodate the non-linear dependencies between covariates and survival implied by the Cox proportional-hazards model, the non-linear terms of the continuous predictors and the interaction terms, we performed imputation using Substantive Model Compatible Fully Conditional Specification [28]. This ensures compatibility between the imputation models and the outcome model. Data were imputed under the most comprehensive outcome model under study.

### 3.4. Hypothesis Testing

We compared models using the multi-parameter pooled Wald-test, which is compatible with multiple imputation [29]. We tested two variants of our main hypothesis that there is a statistical interaction between PMI and PMD: 1. including non-linear interaction terms between PMI and PMD, and stratification per early-stage versus advanced-stage (14 degrees of freedom); 2. only a linear interaction between PMI and PMD without stratification (1 degree of freedom).

### 3.5. Implementation

R version 4.1.0 was used for all statistical analyses. The function ‘smcfcs’ from package smcfcs (version 1.5.0) was used for imputation. Data were imputed using 250 iterations per imputation and 160 fully imputed datasets were generated. The function D1 from package MICE (version 3.13.0) was used for multi-parameter model comparisons. The function rcspline.eval from package Hmisc (version 4.5.0) was used to generate the restricted cubic spline bases for continuous variables. To accommodate the mixed stratification of baseline hazards and hazard ratios we updated the source code of packages smcfcs and survival. The code that implements the imputation and subsequent analysis is publicly available here: https://doi.org/10.5281/zenodo.6107815.

### 3.6. Reporting

For reporting, we adhered to the REMARK statement for biomarker studies [30]. A filled-in form is available in the Appendix A.

## 4. Results

We included 2840 patients and observed 1975 deaths in 5903 patient years. The average age was 68.9 years (standard deviation 10.4, range 32 to 96) and 1692 (59.6%) were male. The median OS since first visit to the radiotherapy department ranged from 3.32 years for stage I patients to 0.53 years for stage IV patients. The baseline characteristics stratified per clinical stage are presented in Table 1 and per-stage Kaplan–Meier survival curves are presented in the Appendix A.

For 1212 of the 2840 patients (42.7%), a PET-CT scan was available within the required time window and muscle measurements were performed. Only one of these PET-CT scans was with intravenous contrast. The median number of days from the scan to the start of follow-up was 33 (interquartile range 21–49). We used 10 variables per patient in the analysis (age, sex, histology group, performance score, BMI, PMI, PMD, clinical stage, survival time, deceased indicator) meaning that there were 28,400 potential values to be recorded. Of these 28,400 values, 21,600 were observed and 6800 were missing, meaning that 76% of the data were available and 24% were imputed. There were 378 patients (13.3%) with no missing values for any of the variables. If a complete-cases analysis were employed, 3780 out of 21,600 (17.5%) available data-points would be used, disregarding 82.5% of the available data.

Three patients with similar PMI but different PMD are presented in Figure 2. There was no statistical evidence for non-linear components and stratification of the interaction terms between PMI and PMD (*p*-value 0.667). Proceeding without non-linear components or stratification of the interaction terms for early-stage versus late-stage, there was clear statistical evidence for a linear interaction between PMI and PMD (hazard ratio 0.915; 95% confidence interval 0.861–0.972; *p*-value 0.004). This confirms the hypothesis that PMI is more positively associated with OS when PMD is higher. In other words, increased psoas muscle area (PMI) is associated with increased OS only when psoas muscle radiodensity (PMD) is sufficiently high. The dependence of the association between PMI and OS on PMD without stratification for early-stage versus late-stage is presented in Figure 3. Again, the shape of the interaction curve is in line with the hypothesis. A table with the parameter estimates for all parameters is presented in the Appendix A.

## 5. Discussion

We conducted a large cohort study of non-small cell lung cancer patients treated with radiotherapy and investigated whether the relationship between muscle quantity and overall survival (OS) depends on muscle radiodensity. We found evidence for a statistical interaction between PMI and PMD for overall survival. Specifically, our experiments confirm the hypothesis that the higher the muscle radiodensity measured in PMD, the more positive the association between muscle quantity measured in PMI and OS. These findings are in line with the general intuition that “having much muscle tissue of good quality (high SMI and SMD)” leads to better overall survival than “having much muscle tissue of lower quality (high SMI but low SMD)”. This statistical interaction provides a potential explanation for the varying results in previous research on muscle mass and OS and has important implications for future research. Future studies on muscle mass and OS should accommodate the statistical interaction between muscle quantity and muscle radiodensity, both in study design and in the analysis. Interaction terms require larger sample sizes to estimate accurately, which should be accounted for in sample size calculations for future studies.

The study cohort consisted of a heterogeneous group of NSCLC cancer patients treated with radiotherapy over a span of 9 years. Stage I and II NSCLC patients treated with radiotherapy are known to have worse overall survival than stage I and II patients treated with surgery because most patients treated with radiotherapy were deemed unfit for surgery. Indeed, the overall survival of stage I and II patients in our population is lower than the general population [32] but similar to other radiotherapy-only populations [33,34]. Given the biological rationale for the hypothesis that muscle quantity of sufficient radiodensity is more protective for OS than muscle quantity that is infiltrated with fatty tissue, we suspect that this hypothesis is true across all cancer types, stages and treatment regimes. The clear statistical evidence in a heterogeneous population of NSCLC cancer patients supports this suspicion, but it will have to be confirmed in future studies in multiple cancer types.

Our study has several limitations. Although our cohort is relatively large, we do not have extensive details on other potential treatments the patients received. In a different study on a partially overlapping cohort of stage III NLSLC patients in our institution, 49 out of 844 (5.8%) patients received surgical treatment [35]. Stage I and II patients receive either surgical treatment or radiotherapy treatment, so the included stage I and II patients were not surgically treated. It is possible that some of the patients with stage IV were treated with surgery for a lower clinical disease stage before inclusion in our cohort, but then later distant metastases were discovered. For these patients, any surgical treatment would have occurred months or years before the start of follow-up in this study. Taken together, this means that the total fraction of patients who had surgical treatment in our cohort is very small and will have a negligible effect on the estimates of our baseline overall survival. Moreover, even if potentially unknown additional treatments like surgery or chemotherapy are important for overall survival, this does not mean that the interaction between muscle quantity, muscle radiodensity and overall survival is different for these treatments, so our main result is likely unaffected. If our aim were to present a new prediction model for use in clinical practice, this lack of detail would be an important limitation. Instead, our goal was to evaluate a hypothesis that has implications for future prediction research. As it is unlikely that the interaction between muscle quantity and muscle radiodensity depends on the given treatment, our findings are meaningful despite the lack of detail regarding other treatments.

During the inclusion period for our study, two different versions of the TNM staging system were used in our center, namely versions six and seven. In our analysis we stratified the baseline hazard per clinical stage. Grouping patients who were staged according to different TNM versions together may have introduced some variation in the baseline survival. Although it is unlikely that this has a significant influence on our main results, we repeated the analysis by stratifying per stage and TNM edition (8 groups in total) as a sensitivity analysis. The results of this sensitivity analysis are in line with our main results: no evidence for non-linear components and stratification of the interaction terms between PMI and PMD (*p*-value 0.758) and clear evidence for a linear interaction between PMI and PMD (hazard ratio 0.918, 95% confidence interval 0.861–0.978, *p* = 0.008).

Finally, there were relatively few patients with complete data. In accordance with statistical guidelines [36,37] we used state-of-the art imputation methods to optimally use the information that was available without excluding any of the patients. In total, 24% of the data points were imputed. Multiple imputation remains valid even when there are many missing values, as long as a sufficient number of imputed datasets is used [38]. Still, future studies should preferably be based on prospective cohorts where important variables are collected in a protocolled manner. Another way to investigate our hypothesis further is by conducting a systematic review of studies on the association between muscle area and muscle radiodensity with OS in cancer patients and performing a meta-regression of the association between muscle area and OS on muscle radiodensity.

We focused our research on body composition measurements. Several studies have highlighted the importance of inflammation, nutritional status and performance score for clinical endpoints in cancer in general and lung cancer specifically [17,39,40]. Due to unavailability of the required measurements, we were unable to study the dependence of the relationship between SMI and SMD and overall survival on measures of nutrition status, such as serum albumin or transferrin, or systemic inflammation, such as the modified Glasgow Prognostic Score [41]. This could be studied in future research.

In conclusion, we found that PMI is more positively associated with overall survival when PMD is higher in a large cohort of NSCLC patients treated with radiotherapy. For accurate and generalizable results, future studies on the relationship between muscle quantity and overall survival in cancer patients should accommodate this statistical interaction in the analysis.

## Figures and Tables

**Figure 1 jpm-12-01191-f001:**
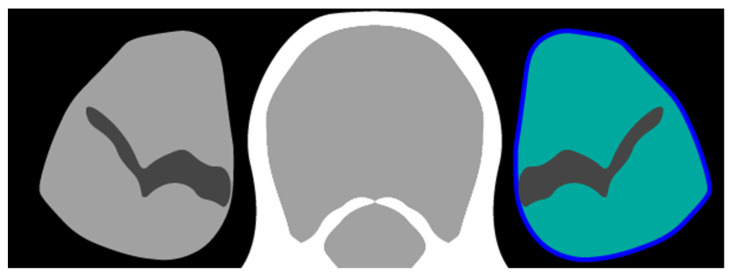
Schematic representation of measurements. The entire area of the psoas muscle on the L3 level was delineated (dark blue circumference). For psoas muscle index (PMI), only voxels with a radiodensity of −30 Hounsfield units (HU) or higher were counted (light green area). For psoas muscle radiodensity (PMD), the average HU of all voxels in the delineated area was calculated, including fatty infiltration of the psoas muscle.

**Figure 2 jpm-12-01191-f002:**
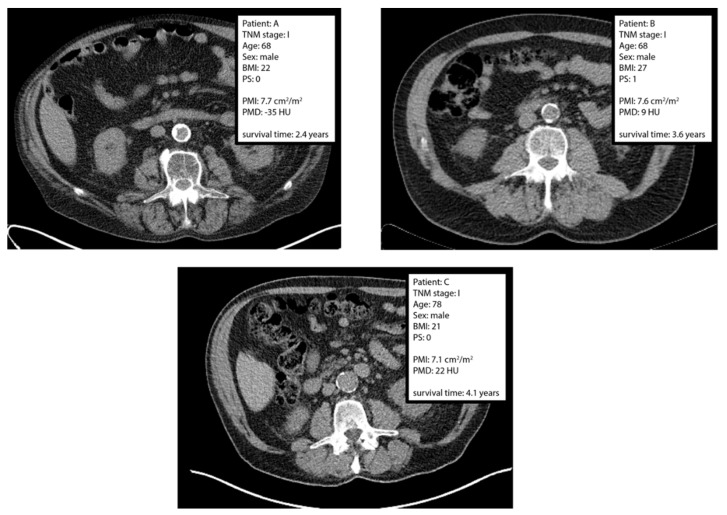
L3-slices of computed tomography scans for three different patients with similar psoas muscle index (PMI) but different psoas muscle radiodensity (PMD). The patients were selected to be similar with respect to stage, age, sex and PMI, but with different PMD. BMI: body mass index. PS: performance score, defined using the Eastern Collaborative Oncology Group standard [23].

**Figure 3 jpm-12-01191-f003:**
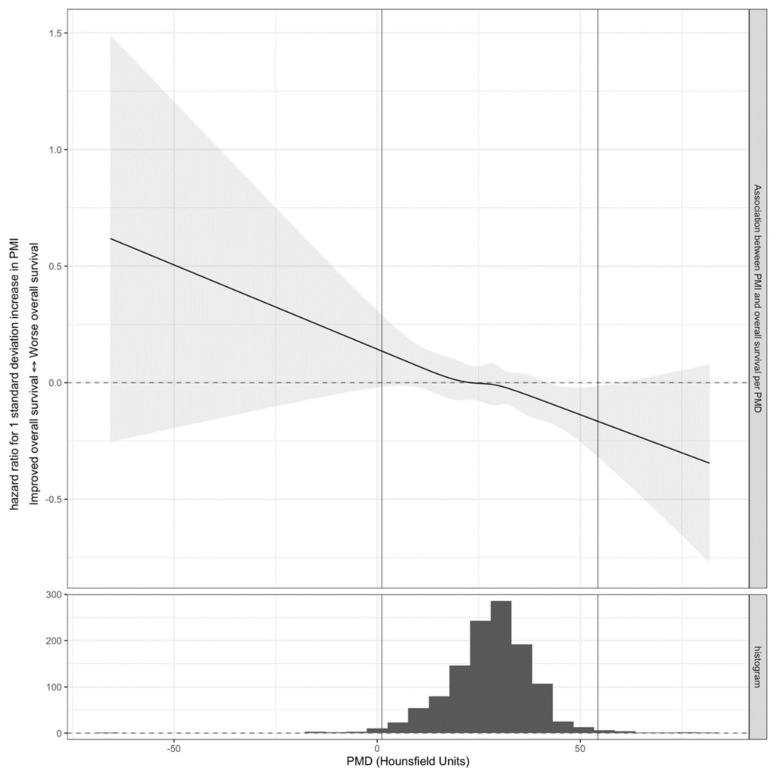
Hazard ratio for a 1 standard deviation increase in PMI for different values of PMD. The average estimate is depicted with a solid black line. The 95% confidence interval is depicted with the gray shaded area. The dashed line indicates the null effect of hazard ratio 1. At the bottom, a histogram for the observed values of PMD is presented. Two vertical lines indicate the region excluding the 1% lowest and 1% highest values of PMD. For this figure, the model was fitted by omitting non-linear terms of PMI and stratification of hazard ratios per early-stage versus advanced-stage; this model does include interaction terms that are non-linear in PMD. The full model also includes interaction terms that are non-linear in PMI, which means that the shape of this interaction function also depends on the value of PMI. To estimate the confidence interval, the model was fitted on 100 bootstrap samples of each of the 160 imputed datasets, following the ‘MI-boot’ procedure [31]. PMI: skeletal muscle index. PMD: skeletal muscle radiodensity.

**Table 1 jpm-12-01191-t001:** Baseline characteristics stratified by clinical disease stage. The mean and standard deviation are calculated based on the non-missing values. The ‘other’ category for histology includes carcinoid tumors, neuro-endocrine tumors and other rare histologic subtypes. Variables age, male sex, SBRT, deceased and survival time had no missing values. A comprehensive dedicated table of radiotherapy targets is presented in the Appendix A. Median overall survival was calculated using the Kaplan–Meier method. PS: performance score, defined using the Eastern Collaborative Oncology Group standard [23]. SD: standard deviation. BMI: body mass index. PMI: psoas muscle mass index. PMD: psoas muscle radiodensity. SBRT: stereotactic body radiation therapy. RT: radiotherapy.

	Overall	Stage I	Stage II	Stage III	Stage IV	Missing
*n*	2840	714	145	871	343	767
age (mean (SD))	68.95 (10.44)	72.65 (9.18)	71.63 (10.47)	66.53 (10.24)	66.26 (10.21)	68.97 (10.73)
male sex (%)	1692 (59.6)	422 (59.1)	89 (61.4)	531 (61.0)	211 (61.5)	439 (57.2)
histology (%)						
adenocarcinoma	595 (21.0)	81 (11.3)	32 (22.1)	272 (31.2)	136 (39.7)	74 (9.6)
no examination	1402 (49.4)	482 (67.5)	55 (37.9)	190 (21.8)	83 (24.2)	592 (77.2)
other	259 (9.1)	46 (6.4)	13 (9.0)	121 (13.9)	56 (16.3)	23 (3.0)
squamous cell	508 (17.9)	74 (10.4)	43 (29.7)	278 (31.9)	59 (17.2)	54 (7.0)
missing	76 (2.7)	31 (4.3)	2 (1.4)	10 (1.1)	9 (2.6)	24 (3.1)
PS (%)						
0	872 (30.7)	177 (24.8)	23 (15.9)	206 (23.7)	64 (18.7)	402 (52.4)
1	553 (19.5)	154 (21.6)	29 (20.0)	239 (27.4)	61 (17.8)	70 (9.1)
>=2	446 (15.7)	102 (14.3)	31 (21.4)	153 (17.6)	80 (23.3)	80 (10.4)
missing	969 (34.1)	281 (39.4)	62 (42.8)	273 (31.3)	138 (40.2)	215 (28.0)
BMI (mean (SD))	25.66 (6.07)	25.57 (5.96)	25.57 (5.26)	25.73 (5.64)	26.42 (7.75)	25.32 (6.23)
BMI missing (%)	1500 (52.8)	309 (43.3)	64 (44.1)	417 (47.9)	212 (61.8)	498 (64.9)
PMI (mean (SD))	6.28 (1.64)	6.27 (1.74)	6.09 (1.41)	6.41 (1.59)	6.21 (1.74)	43.59 (8.43)
PMI missing (%)	1851 (65.2)	386 (54.1)	79 (54.5)	525 (60.3)	266 (77.6)	595 (77.6)
PMD (mean (SD))	27.93 (10.89)	25.81 (12.28)	26.99 (11.32)	30.99 (9.21)	29.33 (10.07)	7.16 (13.88)
PMD missing (%)	1637 (57.6)	314 (44.0)	68 (46.9)	442 (50.7)	262 (76.4)	551 (71.8)
RT target (%)						
lung	1520 (53.5)	667 (93.4)	92 (63.4)	179 (20.6)	126 (36.7)	456 (59.5)
multi-site	1040 (36.6)	29 (4.1)	37 (25.5)	618 (71.0)	146 (42.6)	210 (27.4)
other	114 (4.0)	12 (1.7)	7 (4.8)	16 (1.8)	31 (9.0)	48 (6.3)
mediastinum	97 (3.4)	5 (0.7)	0 (0.0)	43 (4.9)	19 (5.5)	30 (3.9)
hilus	37 (1.3)	0 (0.0)	7 (4.8)	11 (1.3)	5 (1.5)	14 (1.8)
thorax wall	23 (0.8)	1 (0.1)	2 (1.4)	4 (0.5)	8 (2.3)	8 (1.0)
brain	8 (0.3)	0 (0.0)	0 (0.0)	0 (0.0)	7 (2.0)	1 (0.1)
missing	1 (0.0)	0 (0.0)	0 (0.0)	0 (0.0)	1 (0.3)	0 (0.0)
SBRT (%)	1096 (38.6)	643 (90.1)	61 (42.1)	29 (3.3)	39 (11.4)	324 (42.2)
deceased (%)	1975 (69.5)	364 (51.0)	96 (66.2)	674 (77.4)	284 (82.8)	557 (72.6)
survival (median)	1.71	3.32	2.15	1.41	0.53	1.63

## Data Availability

Due to Dutch privacy regulations, the original data for this study cannot be made available. The code that implements the imputation and subsequent analysis is publicly available here: https://doi.org/10.5281/zenodo.6107815.

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
