# Peer review of "The Association between Muscle Quantity and Overall Survival Depends on Muscle Radiodensity: A Cohort Study in Non-Small-Cell Lung Cancer Patients"

_jpm, 2022, doi:10.3390/jpm12071191_

Round 1

Reviewer 1 Report

The authors submitted a paper in which they evaluated the association between muscle radiodensity and quantity and overall survival in patients affected by NSCLC. The hypothesis behind the paper is interesting as the simple CT-measured psoas area might be a very raw measurement of muscle mass that might be hard to generalize due to difference in sex, age, or performance status. The idea to take a “snapshot” of a very heterogeneous population at the time of the diagnosis to verify if a reduction in the quantity and quality of muscle might be used as a surrogate biomarker for survival is intriguing. The Authors deal effectively with the major issues of a very difficult statistical analysis.

Nonetheless, I think that the Authors should deal with some issues:
·        The Authors classified each patient according to the TNM category that was assigned at the time of the diagnosis. However, from the 6th to the 8th edition of TNM the classification was considerably changed (i.e., the T3 class was modified) and probably there is some overlapping between T2-T3 classes. TNM was designed to match patients that most likely will have a similar prognosis and I think that this might be a major flaw of the study. I think that the Authors should classify all the patients according to the most recent version of TNM.
·        The discussion section should be implemented. The authors employed very complex statistical techniques that are not easy to understand for a non-statistician reader and they should make their results more understandable.

Reviewer 2 Report

I would like to congratulate the authors for their interesting and informative paper.

This is an observational cohort study investigating the prognostic significance of muscle mass, as assessed on computed tomography, in patients with lung cancer treated with radiation therapy. Specifically, a deep-learning algorithm was used to measure muscle quantity as psoas muscle index and psoas muscle radiodensity on computed tomography. The study included 2840 patients (60% male) and recorded 1975 deaths. The authors found that psoas muscle index was associated with improved overall survival for higher values of psoas muscle radiodensity (hazard ratio 0.92; 95% confidence interval, 0.86-0.97; P = 0.004). Therefore, the authors conclude that increased muscle quantity is associated with better overall survival when muscle radiodensity is higher in patients with lung cancer treated with radiotherapy.

This is an overall well written study. Here, I have made a few observations and suggestions.

·         The authors stratified the baseline hazard function per clinical stage in four groups (I, II, III, and IV). However, they maintained the TNM version that was used at the time of treatment, namely the sixth, seventh, and eight editions. They may consider commenting on how this may have affected the results.

·         The authors may consider discussing if there were any cases in which radiotherapy was administered after incomplete surgical resection (i.e., R1 or R2 resection).

·         The authors may consider discussing the results of their study in reference to other markers of nutrition, such as serum albumin and transferrin. 

Round 2

Reviewer 1 Report

I would like to thank the authors for addressing my comments.

This manuscript is a resubmission of an earlier submission. The following is a list of the peer review reports and author responses from that submission.

Round 1

Reviewer 1 Report

Dear authors,

Your observational study gives interesting information on the association between muscle quantity and overall survival in patients with NSCLC. However, I have some comments:

As for the methods:

  1. Patient inclusion:
    1. It is not clear what is the cohort entry date. The date of the first visit in a radiotherapy department? Or the date of the first diagnosis of NSCLC?
    2. Despite NSCLC rarely occurs in underage patients, did you included also patients with < 18 years old? Do PMD and SMD values could be influenced by the age of patients? Please motivate this choice.
    3. You stated that you included patients if they have visited the radiotherapy department for “consideration” of treatment. So did you included also those patients who did not receive radiotherapy?
    4. What if a patient had the first visit before 01/2019? Did you excluded him?
    5. Could these patients have received a previous pharmacological treatment?
  2. Definition of exposures and outcomes:
    1. Where did you extract the “non-clinical” variables? Are EHR that you used linked with other external datasources?
    2. Did you have to anonymized the data for the linkage with the Dutch Personal Records Database?
    3. Does patients needed to have at least a minimum time of follow-up?
    4. This chapter is called definition of exposures and outcome: outcome is the OS, but what about the exposure? You did not mention it in the chapter text.
    5. Maybe you can structure this chapter in subchapters: Exposure, outcome and other variables (clinical and non-clinical).
    6. Did you have any information about previous pharmacological therapy? Concomitant therapy? Type of therapy?
    7. Did you have any information about surgery?
  1. Statistical analysis:
    1. “Continuous variables were centered by subtracting the mean before entering the analysis.” This statement is not clear.

As for the results:

  1. Appendix is missing. Please provide it.
  2. RT target: What is the record of RT that you reported in table one? The first?
  3. I strongly suggest to report the number of patients receiving a pharmacological treatment before cohort entry, those who received during follow-up, and those who did not. Also the number of patients undergo surgery should be reported.
  4. I didn’t get the rational of reporting the slices of computed tomography scans for these three patients. How did you select them? I think you should comment more this result.
  5. Figure 3 is referred to what population?
  6. Results of the COX model for all the variables included is not reported.

Reviewer 2 Report

A few more informations on the radiotherapy treatments (dose-/volume-concept etc.) would be helpful (if possible - the authors already stated in the "Discussion"-section that they do not have extensive details).

Round 2

Reviewer 1 Report

Dear authors,

Thank you for carefully answer my suggestions and i understand that muscle quantity and muscle radiodensity is unlikely to depend on the given treatment. However, i'm still concerned about the lack of treatment data (both surgery and pharmacological treatment). In particular surgery could affect strongly the survival of such patients and it is not sure that all the patients received a radiotherapy. Moreover, about ¼ of your cohort has missing a stage information

Author Response

Again, we thank the reviewer for their comments and swift reply.

We agree with the reviewer that surgery can strongly affect the survival of lung cancer patients, especially if radical surgery were possible for stage III patients who received adjuvant or neo-adjuvant (chemo)radiation. However, even if surgery has an important impact on survival, that does not imply that the relationship between psoas muscle index (PMI), psoas muscle radiodensity (PMD) and survival is different in those patients. Specifically, it is highly likely that more muscle volume (high PMI) of sufficient radiodensity (high PMD) is associated with better overall survival both in patients who had surgery and in patients who did not have surgery. This means that the conclusions of our study, which focus on this relationship between PMI and PMD and overall survival, should remain intact, even if surgery is strongly associated with survival. Analogously, as reported in the manuscript, we did not find evidence that the relationship between PMI, PMD and overall survival is different between early-stage and late-stage patients, despite that stage is crucial for overall survival.

In a cohort of stage III patients of the same institution but curated for a different research question and with slightly different inclusion criteria and dates, we found that 56 out of 854 (6.5%) patients were treated with surgery and (chemo)radiotherapy. The study is published here: https://doi.org/10.1038/s41598-022-09775-9, though surgery was an exclusion criterion for this particular study so the number is not reported there. For stage I and stage II patients, surgery will be very rare as patients will get either surgery or radiotherapy but not both and ours is a radiotherapy cohort. For stage IV patients, surgery will be very rare as well. This means that the overall fraction of patients with surgery is likely to be very small in the cohort of our manuscript. Ideally, we would have information on surgery for each patient, but even if we had this information it would be impossible to add interaction terms for PMI, PMD and surgery / do subgroup analyses because there would be too few patients. Moreover, the impact on the final results of any surgical treatments is likely to be negligible.

As said in our discussion, if we were to propose a new prediction model for clinical use, this lack of detail would be a limitation. Given these considerations however we think that our conclusion that the relationship between PMI and overall survival depends on PMD stands firm despite the lack of detail. As also mentioned, we expect this interaction to be present in other cancer settings as well, also in patients with different treatments. Our claim is that future prediction models using muscle quantity should accommodate the interaction between muscle density. We hope the reviewer and editor see the relevance of publishing this work so that future research in this area can accommodate this interaction and make prediction models that are more accurate and more stable across different institutions and subpopulations.